# Evaluation of pre- and in-hospital workflows and time intervals with acute ischemic stroke patients

**Mikko Helander[1], Timo Iirola[1], Pauli Ylikotila[2], Hilla Nordquist[3]\***

1 Department of Emergency Medical Services, Turku University Hospital and University of Turku, Turku, Finland, 2 Department of Neurocenter, Division of Neurology, Section of Cerebrovascular Disorders, Turku University Hospital and University of Turku, Turku, Finland, 3 Department of Healthcare and Emergency Care, South-Eastern Finland University of Applied Sciences, Kotka, Finland

\* hilla.nordquist@xamk.fi

## Abstract

### Objective

Stroke is one of the leading causes of death and disability globally. Rapid recanalization therapy for acute ischemic stroke (AIS) patients is critical for improving outcome. While in-hospital time intervals have decreased and treatment methods have improved over the past decade, pre-hospital time intervals have remained unchanged. This study aims to develop a comprehensive understanding of the prognosis of AIS patients based on pre-hospital and in-hospital data.

### Methods

A retrospective study was conducted in Southwest Finland covering the period of 01/01/2022-31/12/2022. The study included a total of 174 AIS patients who were conveyed by the emergency medical services (EMS) to Turku University Hospital (TUH) and given intravenous thrombolysis (IVT) and/or endovascular treatment (EVT). Pre- and in-hospital care records of AIS patients were analyzed to evaluate workflows, time intervals, and the impacts of these time intervals on patient outcome. Binary logistic regression analysis was utilized to identify predictors of on-scene time (OST) and favorable outcome.

### Results

The median OST for EMS was 19 minutes. Analysis indicated that the scene of the stroke event being an apartment building, barriers to care such as a locked door, vertigo as a symptom, and EMS taking measurements on-scene were predictive of longer OST. Longer OST was observed to negatively impact patient outcome, along with symptom severity and gender. Using the stroke code as the dispatch code and positive FAST signs were associated with shorter OST. In-hospital median door to needle time (DNT) was 14 minutes in the IVT group and 11 minutes in the IVT+EVT group. EMS prenotification was associated with shorter in-hospital time intervals.

**Data availability statement:** Data cannot be shared publicly because data contains potentially identifying and sensitive patient information. Data are available from the research services of the Wellbeing Service County of Southwest Finland, (contact via timo.iirola@varha.fi or utilizing the contact information provided at https://www.varha.fi/fi/tietoa-meista/tieteellinen-tutkimus/tutkimuspalvelut) for researchers who meet the criteria for access to confidential data.

**Funding:** The author(s) received no specific funding for this work.

**Competing interests:** The authors have declared that no competing interests exist.

## Conclusions

The presence of vertigo as a symptom poses challenges to identification by EMS. Pre-hospital OST meets national targets, but EMS workflows could be optimized to reduce OST and thereby positively influence patient outcome. These findings underscore the need for targeted interventions in EMS protocols to improve stroke care outcomes.

## Introduction

Stroke is one of the leading causes of death and disability worldwide. In 2021, there were 12 million new stroke cases globally, with a prevalence of 94 million, 65% of which were ischemic strokes. Between 1990 and 2021, the global stroke burden increased substantially in absolute numbers. However, the slower decline in age-standardized stroke incidence rates observed during 2019–2021 is likely linked to decreased hospital admissions for an acute stroke during the COVID-19 pandemic [1]. As the population ages, the increasing social and economic burden of stroke poses significant challenges for public health services [2].

For acute ischemic stroke (AIS) patients, the goal is to initiate recanalization therapy as quickly as possible. A short time from symptom onset to recanalization improves treatment outcome [3–9]. In-hospital time intervals in treating AIS patients have been reduced by optimizing treatment protocols [9–11]. Even though in-hospital time intervals have decreased and treatment methods have improved over the past decade, pre-hospital time intervals have remained unchanged [12–15].

The pre-hospital time intervals includes the time it takes the patient to seek help, the Emergency Response Center's (ERC) dispatching time, and the Emergency Medical Services' (EMS) response, on-scene (OST), and conveyance times [12,15–17]. The OST constitutes approximately 40% of the total EMS time [12,16,17], representing a substantial portion of the entire process. This is also the variable EMS personnel can most effectively influence. External variables like traffic or weather conditions influence this less [18–19]. However, specific patient symptoms, such as vertigo, present challenges for EMS [34]. As a nonspecific symptom, vertigo often overlaps with benign conditions, complicating EMS personnel's ability to promptly and accurately identify stroke cases [39,40]. By reducing the OST, it is possible to positively influence patient outcome [9,19–21]. Studies have indicated that the OST during pre-hospital care is where future research and interventions should be directed [16,19,20].

The aim of this study was to develop a comprehensive understanding of the prognosis of acute ischemic stroke (AIS) patients based on pre-hospital and in-hospital data from Southwest Finland. The focus was on the time intervals throughout the care pathway, the factors influencing these intervals, the impact of time on patient outcomes, and identifying areas for improvement—particularly in reducing on-scene time during pre-hospital care. The study involved AIS patients who were conveyed by EMS and treated with recanalization therapies. Our research questions were:

1. What constitutes the workflow in the pre- and in-hospital phases?

2. What kind of time intervals occur in the pre- and in-hospital phases?

3. Which factors influence the time intervals?

4. What is the impact of time on patient outcome?

## Materials and methods

A retrospective register study was conducted using the patient registry from the Wellbeing Services County of Southwest Finland, covering the period of 01/01/2022-31/12/2022. The patient

registry includes both pre-hospital and in-hospital care records, and data was collected in 2023. AIS patients who were conveyed by EMS to Turku University Hospital (TUH) and received intravenous thrombolysis (IVT) and/or endovascular treatment (EVT) were included in the study. The guidelines of the STROBE checklist for observational studies were adhered to.

## Setting

In Finland, which has a population of 5.6 million, responsibility for public health care is divided among 21 wellbeing service counties [22]. Each of these regions independently organizes its social and health services, including EMS, in accordance with the framework established in laws and regulations.

The area covered by this research, the Wellbeing Services County of Southwest Finland (known as the Hospital District of Southwest Finland at the time of the study), is among the most populous Wellbeing Regions in the country. TUH serves the 490,000 residents of the region and is the only hospital in the area that provides treatment for AIS patients [23]. Approximately two-thirds of these residents live in urban areas, while the remaining third reside in rural settings. A distinctive feature is the extensive archipelago area, which presents challenges particularly from the perspective of reaching patients.

All of the EMS units in Southwest Finland operate at an advanced level (ALS), with each having at least one ALS-qualified paramedic. ALS-level paramedics typically hold a four-year bachelor's degree from a University of Applied Sciences or have a registered nurse qualification with additional ALS studies. The other person in the EMS unit can be either an ALS or basic-level (BLS) paramedic or a firefighter. The region is also served by a helicopter emergency medical service (HEMS) unit, which brings physician-level care directly to the scene. EMS personnel utilize electronic patient care records (ePCR) to document patient data and manually enter all data into the EMS unit's computer. The system guides the paramedic with predefined questions to gather information from the patient, such as in cases of stroke.

The treatment pathway for AIS patients begins when either the patient or a bystander recognizes symptoms of a stroke and dials the national emergency number, 112. The ERC operator conducts a risk assessment and, based on the result, dispatches an ambulance with one of four priority codes. Codes A and B are high-risk "lights and sirens" missions; for example, acute stroke in a self-reliant person presenting within 9 hours of symptom onset is classified as a B mission, C is urgent but not immediate, and D is non-urgent. The primary responsibility of EMS when attending to an AIS patient is to identify the symptoms and urgently convey the patient to a hospital capable of providing recanalization therapy.

The pre-hospital stroke protocol in the area studied stipulates that only necessary actions should be performed on-scene, such as gathering patient information, assessing functional abilities, and noting medication taken and the onset time of symptoms. If the patient's condition permits, initial vital signs should be measured and an intravenous (IV) connection established during conveyance.

The in-hospital stroke protocol involves several steps after arrival at the Emergency Department (ED). A neurologist conducts an initial National Institutes of Health Stroke Scale (NIHSS) evaluation and a bedside INR (International Normalized Ratio) value is obtained from the patient. Following calculation of the initial NIHSS score, the patient is moved on the ambulance stretcher to the CT/MRI suite, and recanalization treatment is administered if indicated. If a large vessel occlusion is detected with imaging the patient is referred to the angiography suite.

## Description of the data

The study included patients directly conveyed to TUH by EMS and treated with IVT and/ or given EVT for acute care. Patients with in-hospital AIS, conveyed from other facilities,

missing EMS records, or non-EMS arrivals were excluded. Data on patients who received recanalization therapy during the follow-up period of 01/01/2022-31/12/2022 were obtained from the hospital stroke register. This register provided patient information, details of treatment given, in-hospital timestamps, initial NIHSS score, and 3-month modified Rankin Scale (mRS) score, and indicated whether prenotification was given by EMS. Once the patients to be included were identified, their data were collected from ePCRs between 12/10/2023-19/10/2023. Data was collected using a data collection sheet, where information from patient records (ePCRs) was manually gathered. The data collected included EMS mission information, timestamps, barriers to care encountered by EMS, and workflow in both pre-hospital and in-hospital settings (see Supplemental Methods in S1 File). After data collection, the identifying information was removed from the data collection sheet and replaced with an anonymous identification number. There were no minors in the data.

## Statistical methods

Categorical data were reported as frequencies and percentages. The Kolmogorov-Smirnov test was used to assess the normality of continuous variables. Depending on the distribution, continuous variables were reported as medians with interquartile ranges (IQR), defined as the range between the 25th and 75th percentiles for non-normally distributed data, or as means with standard deviations and minimum-maximum values for normally distributed data. The study calculated the time intervals based on timestamps from both the pre- and in-hospital phases. The Mann-Whitney U test was utilized to analyze differences in the median time intervals between groups.

Binary logistic regression analysis was utilized to identify predictors of OST and favorable outcome. The dependent variable for OST was defined using a cut off time of 20 minutes or less, based on the national target in Finland for OST. The dependent variable for a favorable outcome was defined as a 3-month mRS score of 0-2. (see Supplemental Methods in S1 File). For each independent variable entered into the model, the estimated odds ratio (OR) was reported along with 95% confidence intervals (CI) to interpret the effect size and direction of the associations.

Two pseudo R-squared values were reported to evaluate the goodness of fit of the binary logistic regression models: Cox and Snell R Square and Nagelkerke R Square. These values indicate how well the model explains the variation in the dependent variable. Higher values suggest a better fit. While Cox and Snell R Square is limited in that it cannot reach 1, Nagelkerke R Square adjusts for this limitation, allowing the value to range from 0 to 1 for clearer interpretation.

Missing data for the dependent variable (OST) was imputed using the median value of the observed data solely to assess whether the missing data influenced the results. Sensitivity analyses were conducted for both regression models, comparing the outcomes before and after imputation to evaluate the robustness of the findings.

Statistical significance was set at a p-value of $< 0.05$. All statistical analyses were conducted using SPSS version 28. All figures were created by MH using Adobe Illustrator version 28.3.

## Ethics approval and consent to participate

In this study, the principles of research ethics were adhered to, along with the guidelines for good scientific practice [24]. The study was conducted per the Helsinki Declaration and all the appropriate national guidelines. As a retrospective study that utilized registry data, ethical approval from the ethics committee (equivalent to an IRB) was waived following the Guidelines of the Finnish National Board on Research Integrity [25]. According to Finnish

legislation, patient consent is not required for scientific studies that are based solely on registry data and conducted with appropriate research permissions (Data Protection Act, 5.12.2018/1050, 31§). A research permit (T2070/2023) was obtained from the Turku Clinical Research Center. The research data were handled in an appropriate and secure manner and stored on TUH's secure server.

## Results

A total of 213 patients were identified as having received IVT and/or EVT at TUH between 01/01/2022-31/12/2022. After exclusions, 174 were included in the study (Fig 1).

The patient cohort had a median age of 76 (IQR 67, 83). Women comprised 45% of the group. The majority of incidents occurred outside office hours, predominantly in urban detached houses or apartment buildings, and the median distance to the hospital was 14.7 miles (IQR 4.1, 40.6). The ERC used a high priority dispatch code in 86.8% of cases, and a code suggesting stroke in 73% of these (Table 1).

### Composition of the pre- and in-hospital phase workflows

Following the local stroke protocol, paramedics collected key patient background information in most cases, including previous medical conditions (79.3%), medications (78.7%), and functional abilities (64.5%).

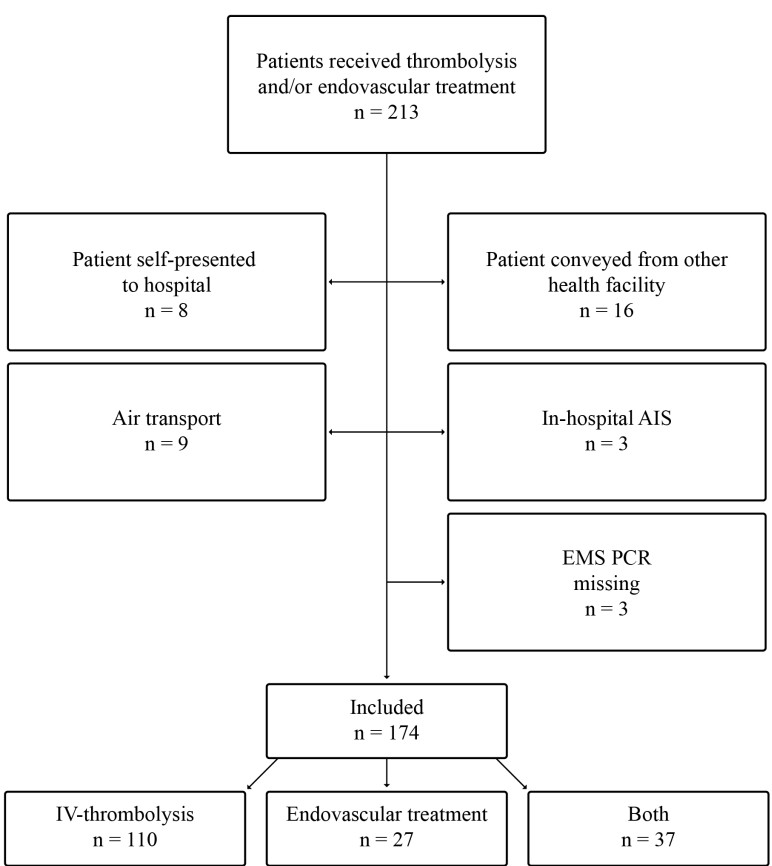

**Fig 1. Flow chart of the study.**

**Table 1. Characteristics of the 174 AIS patients treated with recanalization therapy.**

|  | n/information available | % | Median (IQR) | Min-Max |
|---|---|---|---|---|
| Age, years | 174 |  | 75 (66, 81) | 30–97 |
| Women | 78 | 44.8 |  |  |
| Monday-Friday | 122 | 70.1 |  |  |
| Office hours[1] | 59 | 33.9 |  |  |
| Wintertime[2] | 42 | 24.1 |  |  |
| Scene of the stroke event |  |  |  |  |
| Detached house | 68 | 39.1 |  |  |
| Apartment building | 65 | 37.4 |  |  |
| Rowhouse | 11 | 6.3 |  |  |
| Other[3] | 30 | 17.2 |  |  |
| Urban area | 138 | 79.3 |  |  |
| Distance to TUH[4], miles | 174 |  | 15 (4, 41) | 1–69 |
| Dispatch code |  |  |  |  |
| Stroke code[5] | 127 | 73 |  |  |
| High priority code | 151 | 86.8 |  |  |
| Conveyance code |  |  |  |  |
| Stroke code | 148 | 85.1 |  |  |
| High priority code | 161 | 92.5 |  |  |
| Low to high[6] | 16 | 9.2 |  |  |
| High to low[6] | 7 | 4.0 |  |  |
| EMS requested care instructions from on-call physician | 57 | 32.8 |  |  |
| Physician at scene | 2 | 1.1 |  |  |
| EMS prenotification | 156 | 89.7 |  |  |
| FAST[7] positive[8] | 150 | 86.2 |  |  |
| Initial NIHSS[9] scores | 164 | 94.3 | 7 (3, 12) | 0–25 |
| Recanalization treatment groups |  |  |  |  |
| Thrombolysis | 110 | 63.2 |  |  |
| Thrombectomy | 27 | 15.5 |  |  |
| Both | 37 | 21.3 |  |  |
| 3-months mRS[10], scores | 171 |  | 2 (0, 3) | 0–6 |

[1]Monday to Friday, 8 AM to 4 PM.

[2]Thermal winter in Southwest Finland, 01/01/2022-16/03/2022 and 16/11/2022-31/12/2022.

[3]Outdoors, public space, assisted living facility, industrial property, housing service, or accommodation unit.

[4]Turku University Hospital.

[5]High priority stroke code.

[6]After the on-scene evaluation, the paramedics used either a lower or higher priority conveyance code compared to the dispatch code.

[7]FAST (Face, Arm, Speech, Time) stroke recognition algorithm.

[8]FAST-positive patients have one or more of these symptoms: Face drooping, Arm weakness, Speech difficulties.

[9]National Institutes of Health Stroke Scale.

[10]Modified Rankin Scale.

During their examinations, paramedics most commonly assessed arm motor function (95%), speech (94%), and facial motor function (83%). The most frequent findings were speech disturbances (71%), arm weakness (63%), leg weakness (50%), and facial drop (41%). A complete overview of the examinations and findings is provided in S1 Table in S1 File.

The initial vital signs measured by paramedics and the timing of the measurement – whether on-scene or during conveyance – are presented in S2 Table in S1 File. Details of all measurements, procedures, and treatments are presented in S3 Table in S1 File.

EMS personnel requested care instructions from on-call physicians in 32.8% of cases, and a physician was present at the scene in 1.1% of the cases. After the on-scene evaluation, paramedics upgraded the priority code in 9.2% of cases and downgraded it in 4% of cases. In 92.5% of cases, paramedics used a high-priority conveyance code, with 85.1% of cases being assigned a stroke code. The initial NIHSS score assigned in the emergency department (ED) and details of treatments administered are presented in Table 1.

## Pre- and in-hospital time intervals

The study identified a total of 12 different time intervals from onset of symptoms to the patient receiving treatment (Fig 2 and S4 Table in S1 File). The total time taken by EMS from dispatch to arriving at the hospital door was 57 minutes (IQR 43, 77.5), this includes handover time. The in-hospital time interval from door to needle (DNT) regarding IV-thrombolysis was 14.5 minutes for the IVT group (IQR 9.75, 27.25) and 11 minutes for the IVT+EVT group (IQR 8, 17.5).

## Variables associated with on-scene time

The scene of the stroke event being an urban location or apartment building, barriers to care encountered by EMS, vertigo as a symptom, EMS requesting care instructions, and the performance of on-scene procedures such as 12-lead ECG, IV access, and blood pressure measurements were associated with increased OST (Table 2). In contrast, dispatching with a stroke code and the presence of positive FAST signs were associated with reduced OST.

Binary logistic regression analysis was used to find predictors that affect OST. In the first model, measuring blood pressure on-scene was the most significant factor, reducing the chances of keeping OST under 20 minutes by 84%. As more factors were added in the next models, stroke events occurring in apartment buildings and barriers to care further lowered the chances of a shorter OST. In the final model, vertigo was found to have the strongest effect, reducing the likelihood of an OST under 20 minutes by 97%. All the factors identified made it less likely to meet the 20-minute OST target (Table 3). In Table 3, the Cox and Snell $R$ square value is 0.306 and the Nagelkerke $R$ square value is 0.410, indicating that the model explained between 30.6% and 41% of the variance in the dependent variable. Sensitivity analyses using the imputation of missing values (OST) confirmed that the observed associations, including the timing of blood pressure measurement and the presence of barriers to care, remained robust, indicating that the missing data did not significantly impact on the results of the regression models.

## Association of EMS prenotification with in-hospital time intervals

Prenotification was given by EMS to the hospital in 89.7% (n = 156) of cases and was associated with a decreased in-hospital time intervals for patients who received IVT (Table 4). For patients who received EVT, the difference in time intervals was not statistically significant.

## Variables associated with patient outcome

Binary logistic regression analysis was used to identify factors that predict patient outcomes (Table 5). In the first model for the IVT group, the initial NIHSS score was found to be the strongest predictor. Each additional point on the NIHSS scale reduced the likelihood of a favorable outcome by 22.5% (OR 0.775, 95% CI: 0.68–0.88). This trend remained in the final

## Pre-hospital phase

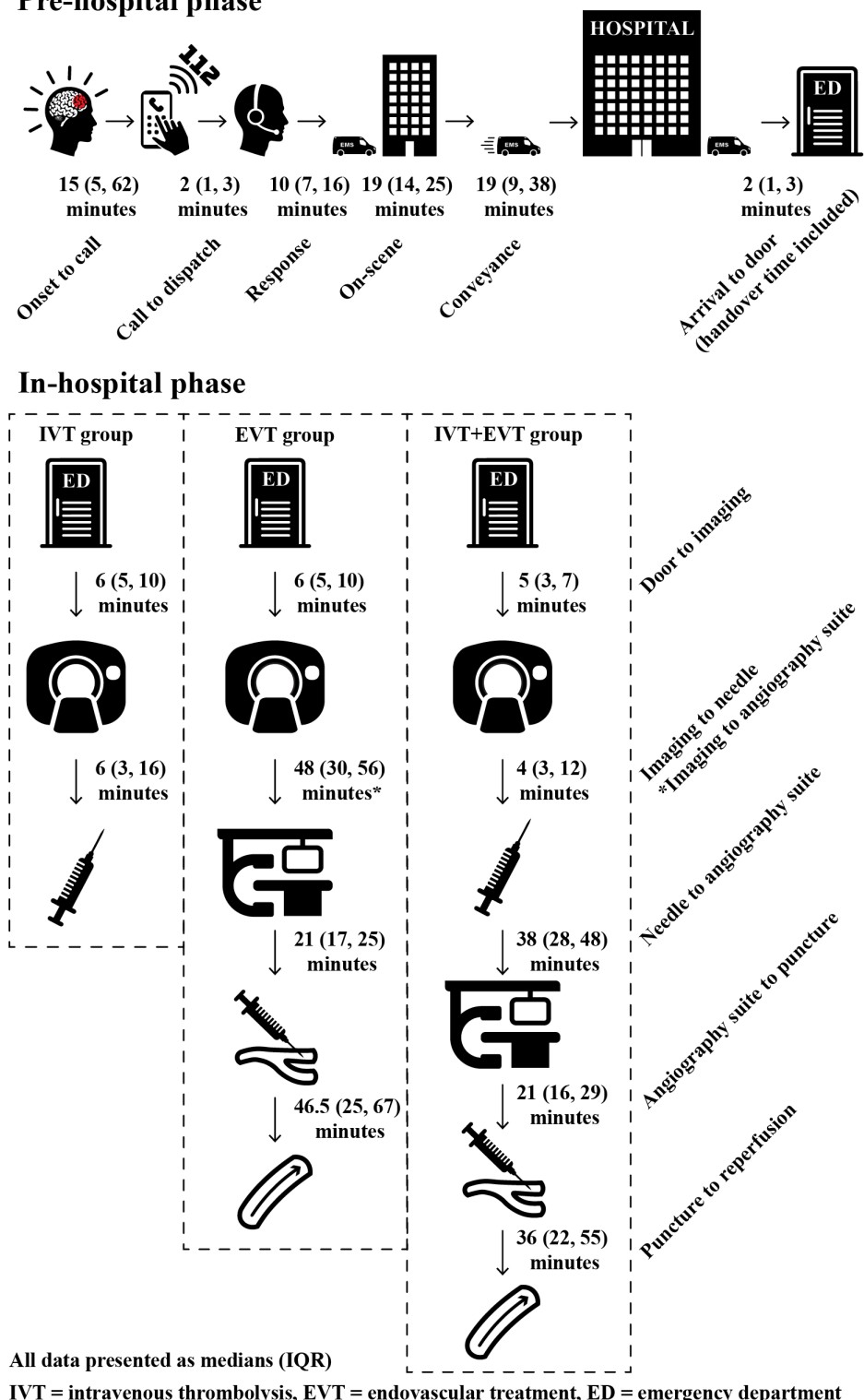

**Fig 2. Time intervals in pre- and in-hospital phases.**

**Table 2. Variables associated with on-scene time, minutes.**

| | Median (IQR) | P-value |
|---|---|---|
| Dispatch: Stroke code[1] | | <0.001 |
| Yes | 18 (13.75, 24) | |
| No | 24 (17, 32.25) | |
| Urban area | | 0.013 |
| Yes | 19 (15, 26) | |
| No | 14 (11, 23.5) | |
| Scene of the stroke event: Apartment building | | <0.001 |
| Yes | 23 (17.25, 27) | |
| No | 17 (13, 23.75) | |
| Barriers to care | | <0.001 |
| Yes | 26.5 (21.75, 38.5) | |
| No | 18 (13.75, 24) | |
| FAST-positive[2] | | 0.002 |
| Yes | 18 (14, 24) | |
| No | 25 (18.5, 38) | |
| Vertigo as symptom | | 0.004 |
| Yes | 26 (23, 43) | |
| No | 18 (14, 25) | |
| 12-lead ECG[3] on-scene | | 0.003 |
| Yes | 35.5 (31, 49) | |
| No | 18.5 (14, 25) | |
| IV access[4] on-scene | | 0.024 |
| Yes | 21 (15, 26) | |
| No | 17 (13, 23.5) | |
| Initial blood pressure measured on-scene | | <0.001 |
| Yes | 22 (16.5, 27) | |
| No | 14 (12, 18) | |
| EMS[5] requested care instructions | | <0.001 |
| Yes | 25 (17, 30) | |
| No | 17 (13.5, 22.5) | |

[1]High priority stroke code.

[2]FAST-positive patients have one or more of these symptoms: Face drooping, Arm weakness, Speech difficulties.

[3]12-lead electrocardiogram.

[4]Intravenous access obtained.

[5]Emergency Medical Services.

model, where the NIHSS score was still the most significant factor, with each additional point lowering the chances of a favorable outcome by 26% (OR 0.74, 95% CI: 0.64-0.86).

In addition, OST emerged as a significant predictor in the final model for the IVT group. A longer on-scene time slightly reduced the likelihood of a favorable outcome, with an OR of 0.948 (95% CI: 0.9–0.998), meaning that for each additional minute spent on-scene, the chances of a favorable outcome decreased by approximately 5%.

In the IVT+EVT group, the initial NIHSS score was again the most important predictor, with each additional point reducing the chances of a favorable outcome by 11.4% (OR 0.886, 95% CI: 0.785–1.000).

**Table 3. Logistic regression analysis: Models' predictions of on-scene time.**

| | Odds Ratio (95% confidence interval) | P-value |
|---|---|---|
| Model 1: | | |
| Initial blood pressure measured on-scene | 0.16 (0.07 - 0.37) | < 0.001 |
| Model 2: | | |
| Scene of the stroke event: Apartment building | 0.3 (0.14 - 0.63) | 0.002 |
| Initial blood pressure measured on-scene | 0.17 (0.07 - 0.41) | < 0.001 |
| Model 3: | | |
| Scene of the stroke event: Apartment building | 0.27 (0.12 - 0.60) | 0.001 |
| Barriers to care | 0.155 (0.05 - 0.53) | 0.003 |
| Initial blood pressure measured on-scene | 0.213 (0.09 - 0.51) | < 0.001 |
| Model 4 | | |
| Scene of the stroke event: Apartment building | 0.23 (0.1 - 0.53) | < 0.001 |
| Barriers to care | 0.14 (0.04 - 0.51) | 0.003 |
| Vertigo as symptom | 0.03 (0.00 - 0.37) | 0.006 |
| Initial blood pressure measured on-scene | 0.17 (0.07 - 0.44) | < 0.001 |

**Table 4. Association of prenotification with in-hospital time intervals, minutes.**

| | n/information available | Median (IQR) | P-value |
|---|---|---|---|
| Door to imaging | | | <0.001 |
| Prenotification | 156 | 6 (4, 8) | |
| No prenotification | 18 | 28.5 (12.25, 47.25) | |
| Door to needle | | | <0.001 |
| Prenotification | 132 | 11 (9, 20) | |
| No prenotification | 15 | 37 (20, 63) | |
| Door to puncture | | | 0.203 |
| Prenotification | 61 | 75 (57, 88.5) | |
| No prenotification | 4 | 96.5 (69, 239.5) | |

In Table 5, the final models show Cox and Snell R square values of 0.296 for the IVT group and 0.137 for the IVT+EVT group. The Nagelkerke $R$ square values are 0.415 for the IVT group and 0.186 for the IVT+EVT group, indicating that the models explained between 29.6% and 41.5% of the variance in the dependent variable for the IVT group and between 13.7% and 18.6% for the IVT+EVT group. Sensitivity analyses conducted with imputed data demonstrated that the main predictors, including the initial NIHSS score and OST, remained consistent, suggesting that the missing data did not materially alter the outcomes.

Overall, both the initial NIHSS score and on-scene time were key predictors of patient outcomes, with higher stroke severity and longer on-scene times being associated with a lower probability of favorable recovery.

## Discussion

Numerous factors contribute to time intervals in the treatment pathway for AIS patients, but the ability to accurately recognize AIS symptoms at each stage is crucial. Our findings demonstrate that the likelihood of a favorable outcome decreases with every additional minute from onset to treatment, consistent with previous studies [3–9,21,26].

**Table 5. First part of logistic regression analysis: Models' predictions of outcome.**

| Patient group | n | Odds Ratio (95% confidence interval) | P-value |
|---|---|---|---|
| IVT[1] | 85 | | |
| Model 1: | | | |
| Initial NIHSS[2] score | | 0.775 (0.679 – 0.885) | < 0.001 |
| Model 2: | | | |
| Initial NIHSS score | | 0.77 (0.667 – 0.883) | <0.001 |
| Gender, women | | 0.289 (0.095 – 0.876) | 0.028 |
| Model 3: | | | |
| Initial NIHSS score | | 0.744 (0.641 – 0.863) | <0.001 |
| Gender, women | | 0.282 (0.089 – 0.895) | 0.032 |
| On-scene time | | 0.948 (0.9 – 0.998) | 0.043 |
| IVT+EVT[3] | 30 | | |
| Model 1: | | | |
| Initial NIHSS score | | 0.886 (0.785 – 1.000) | 0.05 |

[1]Intravenous thrombolysis.

[2]National Institutes of Health Stroke Scale.

[3]Endovascular treatment. A favorable outcome was defined as a 3-month Modified Rankin Scale (mRS) score of 0-2. Missing variables were removed in the analysis.

Symptom recognition is influenced by the patient's knowledge and the presence of bystanders [2,27]. Poor recognition of stroke symptoms leads to delay in access to treatment. Yet, in our study, the median onset-to-call time was 15 minutes, suggesting that patients and bystanders were relatively effective in recognizing stroke symptoms. This is in line with a previous Finnish study [27]. For comparison, a recent Danish study reported a median onset-to-call time of 27 minutes [28], highlighting that delays in seeking help remain a key barrier to timely access to recanalization therapy.

The ability of the ERC to identify AIS patients plays a key role in EMS time intervals. Not only does it affect response time, but it may also influence the paramedics' readiness and decisions on-scene. In this study, the OST was 18 minutes when the ERC dispatched paramedics using a stroke code, compared to 24 minutes with another code. Previous reports show similar results, with OST ranging from 17 to 23 minutes [17,29]. This suggests that clear symptoms during the emergency call expedite on-scene decision-making. Our findings show that the ERC correctly identified the majority of cases, assigning high-priority dispatch codes in 87% of missions and using the stroke code in 73%, which is in line with previous studies [29–33].

EMS also identified most patients in need of recanalization therapy, using the stroke code for 85% of patients. This is consistent with previous Finnish studies, where high-priority conveyance codes were used in 87–93% of cases [29,31]. In the United States, a recent cohort study reported an EMS stroke recognition sensitivity of 73.5% [34], while another Finnish study found that paramedics' preliminary diagnoses of stroke had 81% accuracy compared to discharge diagnoses [35]. However, it should be noted that this study focused on stroke recognition sensitivity and does not provide a fully comprehensive assessment of the EMS system's overall performance, as specificity was not measured.

The workflow of EMS upon encountering a patient generally followed the study area's stroke protocol, which is based on Finnish guidelines [3,36]. In nearly all cases, paramedics conducted the FAST stroke screen during the initial assessment, and the proportion of FAST-positive cases in this study (78–87%) was consistent with other reports [29,31,37]. Patients with positive FAST signs experienced shorter OST (18 minutes vs. 25 minutes),

highlighting that clear symptoms lead to quicker decision-making. The use of stroke screening tools, recommended in several guidelines [3,8,38], is well supported by our results and should continue to be a priority in stroke management.

In 7% of cases where patients were not conveyed to the hospital with high priority, care instructions were sought from the on-call physician in over 50% of those cases. These patients typically presented with milder strokes. EMS requested care instructions in 33% of cases overall, which, despite the stroke protocol advising direct conveyance to the hospital when stroke symptoms are clear, suggests that paramedics may have been uncertain when symptoms were unclear or minor. Increased OST when care instructions were requested also indicates that minor symptoms led to delays in initiating conveyance.

Urban locations, particularly apartment buildings, and barriers to care were associated with longer OST in AIS cases. Approximately 80% of the cases occurred in urban areas, with 37% in apartment buildings. Barriers to care were reported in 16% of missions, the most common being patient access issues (34.4%) and difficulties in assessment and management (34.4%). These barriers often resulted from locked doors, difficult terrain, or challenging environments such as the archipelago, delaying paramedics' ability to reach and manage the patient.

Certain conditions, such as confusion, severe vomiting, and unconsciousness, required immediate interventions on-scene, including airway management, IV access, and 12-lead ECGs, contributing to prolonged OST.

Vertigo, in particular, was identified as a challenge that notably increased OST. Vertigo has also been associated with strokes that are often missed by EMS [34], making it a diagnostic challenge [39–40].

Overall, OST constituted approximately one-third of the total EMS time, consistent with previous studies [16,41]. The median OST in our study was 19 minutes, comparable to other Nordic studies [13,18,31,41]. Efforts to reduce OST could include collecting data en route, enhancing training [18], and establishing a maximum OST target [42]. Time management in EMS can be improved through effective en route preparation, thorough paramedic protocol training, and efficient team communication. By utilizing en route information to form a situational picture and ensure seamless access, paramedics can streamline operations. Efficiency and decision-making can be further enhanced by emphasizing time-criticality in protocols, prioritizing essential on-scene actions, incorporating the technical deconstruction of tasks, and developing a feedback system to drive continuous improvement. The national target OST in Finland is less than 20 minutes, while the American Heart Association (AHA) recommends less than 15 minutes [38,43]. In this study, the national target was achieved in 57% of cases, and the AHA's 15-minute target was met in 32% of cases.

Prenotification to the hospital occurred in nearly 90% of the cases, consistent with another Finnish study [41]. The AHA registry reports prenotification in 67% of conveyed stroke patients [6]. However, our study focused only on true positive stroke patients, and therefore, the results cannot be fully compared.

Prenotification has been shown to reduce in-hospital time intervals, speeding up imaging and treatment times. Our findings align with previous studies [44,45], demonstrating that prenotification shortens the door-to-imaging and door-to-needle times. Compared with these studies, the reductions observed in our research were more pronounced, likely due to a smaller sample size. Monitoring prenotification rates is also recommended as a quality indicator in stroke care guidelines [8,36,38,43,46]. In the future, it could be valuable to study at what point during the EMS mission the prenotification is made and how it affects the time intervals.

The AHA's primary goal is for 85% of stroke patients to receive intravenous thrombolysis (IVT) within 60 minutes of arrival, with a secondary goal of 50% receiving treatment within

30 minutes [45]. In our study, over 93% of IVT patients received treatment within 60 minutes, and 97% of IVT+EVT patients met this target. Furthermore, 81% of IVT and 95% of EVT patients were treated within 30 minutes. The median DNT was 14.5 minutes for the IVT group and 11 minutes for the IVT+EVT group, indicating that our hospital met the AHA goals. However, while recanalization therapy is provided quickly, our findings suggest that further improvements are possible, particularly in the pre-hospital phase. Future challenges include addressing the demands of an aging population and access issues related to urbanization.

## Limitations

The study has limitations due to its retrospective design, which affected data accuracy and completeness. Missing or erroneous timestamps in the pre-hospital phase may have influenced the accuracy of the time interval assessments. To mitigate this uncertainty, timestamps were cross-verified against other records—such as EMS logs, hospital arrival times, and conveyance distances—to identify and exclude implausible data points. Missing data for OST was addressed using median imputation. Sensitivity analysis comparing complete cases and imputed datasets showed that the imputations had little effect on the results, suggesting that minimal uncertainty was introduced by the missing data. Despite these measures, it is acknowledged that the missing data may have affected the reliability of the results.

The focus on Southwest Finland may limit the generalizability of findings to other regions or countries. Manual data extraction introduces human error risks. The small sample size reduces the study's reliability and generalizability, limiting the ability to detect differences or associations that might exist in a larger population. In the study, the onset time for symptoms was determined with a 30-minute accuracy regarding time intervals. There were 33 wake-up strokes, and in 38 cases, the onset time could not be determined within 30 minutes. Paramedics recorded the last known well time or the time of symptom onset, but these were further clarified in the hospital. Onset times that could not be determined were excluded from the study, which may limit the reliability of the research. Additionally, while the study focused on the sensitivity of the system in identifying AIS cases, it did not assess specificity or the potential for overtriage, which could result in unnecessary resource use, such as the overuse of CT/MRI scans and EMS. This focus on sensitivity without considering specificity may limit the comprehensive evaluation of the system's overall performance.

## Conclusion

AIS patients were generally well-identified throughout the phases of the care chain; however, without assessing false positives, it is difficult to fully evaluate the accuracy of this identification. The presence of vertigo as a symptom poses challenges to identification by EMS. EMS OST median meets national targets, but EMS workflows could be optimized to further reduce OST and thereby positively influence patient outcome. EMS prenotification decreases in-hospital time intervals. Our recommendations for reducing OST include en route data collection, streamlining workflows through education, and setting a target time for OST in the protocol. Further research is required on the impact of en-route data collection on OST.

## Supporting information

**S1 File. Supporting information.**
(DOCX)

## Acknowledgments

MH would like to express special thanks to Jouko Huuskonen for making this writing process possible.

## Author contributions

**Conceptualization:** Mikko Helander, Timo Iirola, Pauli Ylikotila, Hilla Nordquist.

**Data curation:** Mikko Helander, Pauli Ylikotila.

**Formal analysis:** Mikko Helander.

**Project administration:** Mikko Helander, Timo Iirola, Pauli Ylikotila, Hilla Nordquist.

**Resources:** Mikko Helander, Timo Iirola, Pauli Ylikotila, Hilla Nordquist.

**Supervision:** Timo Iirola, Pauli Ylikotila, Hilla Nordquist.

**Visualization:** Mikko Helander.

**Writing – original draft:** Mikko Helander.

**Writing – review & editing:** Mikko Helander, Timo Iirola, Pauli Ylikotila, Hilla Nordquist.

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
