## [Decision Letter · Decision Letter 0]

2 Jan 2025

PONE-D-24-43099Evaluation of pre- and in-hospital workflows and time intervals with acute ischemic stroke patientsPLOS ONE

Dear Dr. Nordquist,

Thank you for submitting your manuscript to PLOS ONE. After careful consideration, we feel that it has merit but does not fully meet PLOS ONE’s publication criteria as it currently stands. Therefore, we invite you to submit a revised version of the manuscript that addresses the points raised during the review process.

We look forward to receiving your revised manuscript.

Kind regards,

Cem Bilgin

Academic Editor

PLOS ONE

Journal Requirements:

2. We note that you have indicated that there are restrictions to data sharing for this study. For studies involving human research participant data or other sensitive data, we encourage authors to share de-identified or anonymized data. However, when data cannot be publicly shared for ethical reasons, we allow authors to make their data sets available upon request. For information on unacceptable data access restrictions, please see http://journals.plos.org/plosone/s/data-availability#loc-unacceptable-data-access-restrictions .

b) If there are no restrictions, please upload the minimal anonymized data set necessary to replicate your study findings to a stable, public repository and provide us with the relevant URLs, DOIs, or accession numbers. Please see http://www.bmj.com/content/340/bmj.c181.long for guidelines on how to de-identify and prepare clinical data for publication. For a list of recommended repositories, please see https://journals.plos.org/plosone/s/recommended-repositories . You also have the option of uploading the data as Supporting Information files, but we would recommend depositing data directly to a data repository if possible.

Reviewers' comments:

Reviewer's Responses to Questions

**Comments to the Author**

1. Is the manuscript technically sound, and do the data support the conclusions?

Reviewer #1: Yes

Reviewer #2: Yes

2. Has the statistical analysis been performed appropriately and rigorously? 

Reviewer #1: N/A

Reviewer #2: Yes

3. Have the authors made all data underlying the findings in their manuscript fully available?

Reviewer #1: Yes

Reviewer #2: Yes

4. Is the manuscript presented in an intelligible fashion and written in standard English?

Reviewer #1: Yes

Reviewer #2: Yes

5. Review Comments to the Author

Reviewer #1: This article is about the prognosis of acute ischemic stroke (AIS) patients based on pre-hospital and in-hospital data, focusing on time intervals and their impact on patient outcomes. It analyzes workflows, identifies factors influencing on-scene time (OST), and evaluates strategies to optimize pre-hospital care for better outcomes. Longer OSTs were associated with worse patient outcomes, emphasizing the critical importance of minimizing delays during pre-hospital care(As expected). Factors such as symptom severity, the presence of locked doors, and gender differences significantly influenced OST.

However, I couldn't find any novelty on your article. Please my comments below;

Abstract:

- The objectives are clear, but consider adding a brief sentence highlighting the significance of the findings.

Introduction:

-Consider explaining why vertigo poses a particular challenge for EMS earlier in the introduction.

Methods:

- Article needs detailed and transparent methodology. However, the inclusion and exclusion criteria can be summarized more concisely.

- Please give statistical results in Results part not in methods

Discussion:

- I suggest expanding on practical solutions to reduce OST based on your findings, such as EMS training or community interventions.

- A paragraph could be added about "How do your findings compare with other studies focusing on EMS prenotification and its impact on in-hospital workflows?"

Reviewer #2: This retrospective study, conducted in Southwest Finland, evaluated the impact of pre-hospital and in-hospital time intervals on outcomes for 174 acute ischemic stroke (AIS) patients. The findings indicated that the median on-scene time (OST) for emergency medical services (EMS) was 19 minutes, with longer OST associated with poorer outcomes. Key factors influencing OST included the stroke event location and specific patient symptoms. The study also highlighted the benefits of EMS prenotification in reducing in-hospital time intervals, with median door-to-needle times of 14 minutes for intravenous thrombolysis (IVT) and 11 minutes for IVT combined with endovascular treatment (EVT).

After addressing the concerns below point by point, the manuscript can be considered for publication.

1) It would be beneficial to include more recent statistics than those from 2019. The authors could also make a brief comparison between global incidence and prevalence over the past five to ten years, particularly considering the impact of the COVID-19 outbreak.

2) Additionally, it would be helpful if the authors could identify uncertainties in time measurements. For instance, they mention in the Limitations section of the manuscript that "Missing or erroneous timestamps in the pre-hospital phase may have influenced time interval and outcome assessments." It would be important to clarify how the authors can quantify the uncertainty related to these missing or erroneous timestamps.

3) Furthermore, if the authors have access to patients' pathological results for blood clots, could they establish a correlation between treatment time and treatment success? Although this may not be the main focus of the manuscript, patient-specific characteristics of blood clots could significantly influence treatment outcomes.

4) Lastly, it would be valuable if the authors could provide future predictions based on the factors affecting the time it takes to transport patients to the hospital.

6. PLOS authors have the option to publish the peer review history of their article (what does this mean? ). If published, this will include your full peer review and any attached files.

**Do you want your identity to be public for this peer review?** For information about this choice, including consent withdrawal, please see our Privacy Policy .

Reviewer #1: No

Reviewer #2: No

---

## [Author Response · Author response to Decision Letter 1]

7 Feb 2025

Please see our response letter (attached).

---

## [Editor Report · Decision Letter 1]

9 Feb 2025

Evaluation of pre- and in-hospital workflows and time intervals with acute ischemic stroke patients

PONE-D-24-43099R1

Dear Dr. Nordquist,

We’re pleased to inform you that your manuscript has been judged scientifically suitable for publication and will be formally accepted for publication once it meets all outstanding technical requirements.

Kind regards,

Cem Bilgin

Academic Editor

PLOS ONE
---

## [Editor Report · Acceptance letter]

PONE-D-24-43099R1

PLOS ONE

Dear Dr. Nordquist,

I'm pleased to inform you that your manuscript has been deemed suitable for publication in PLOS ONE. Congratulations! Your manuscript is now being handed over to our production team.

Kind regards,

on behalf of

Dr. Cem Bilgin

Academic Editor

PLOS ONE